# Processing Strategies for Extraction and Concentration of Bitter Acids and Polyphenols from Brewing By-Products: A Comprehensive Review

**Klycia Fidélis Cerqueira e Silva** [1,*], **Monique Martins Strieder** [2], **Mariana Barreto Carvalhal Pinto** [1], **Maurício Ariel Rostagno** [2] **and Miriam Dupas Hubinger** [1,*]

1 Department of Food Engineering and Technology (DETA), School of Food Engineering, University of Campinas (UNICAMP), Campinas 13083-862, Brazil
2 School of Applied Sciences (FCA), University of Campinas (UNICAMP), Limeira 13484-350, Brazil
* Correspondence: klycia.fidelis@gmail.com (K.F.C.e.S.); mhub@unicamp.br (M.D.H.); Tel.: +55-(19)-3521-4036 (M.D.H.)

**Abstract:** Annually, 221 million tons of agro-food by-products are generated worldwide, causing diverse environmental issues due to incorrect discharge. Hot trub, spent hops, brewer's spent grains, and brewer's spent yeast are the by-products produced in the beer manufacturing chain. These by-products contain fibers, proteins, polyphenols, essential oils, and taste compounds, presenting high possibilities of use as alternative raw materials. In this review, we compiled the knowledge gaps of brewing by-product reuse, from phytochemical compound extractions to concentration approaches, mainly concerning bitter acids and polyphenols. Moreover, we assessed and discussed the emerging technologies and alternative solvents that have allowed for higher extraction yields. We illustrated the importance of purification and concentration steps of non-destructive methods for added value in products from reuse approaches. Finally, we showed the relevance of scale-up and economic feasibility studies in order to encourage the implementation of facilities that produce bitter acids and polyphenols from alternative sources such as hot trub and spent hops.

**Keywords:** hops bitter acids; beer waste; phenolic compounds; xanthohumol; emerging technologies; deep eutectic solvent; hops by-products

## 1. Introduction

It is estimated that 221 million tons/year of agro-food by-products are generated worldwide, considering the production and retail chain [1]. Agro-food by-product uses have been expanding as an approach to sustainably managing food production due to climate changes and the growing global population [2]. The brewing sector contributes heavily to this residue generation since beer is the world's most consumed alcoholic beverage. A total of 1.86 billion hectoliters of beer were produced worldwide just in 2021 [3]. Mathias et al. [4] estimated that for every hectoliter of beer produced, an amount between 15.7 and 23.4 kg of solid by-products is generated. Thus, in 2021, a volume of 4,352,000 tons was produced of solid by-products from the brewing industry. These solid by-products are generated during the stages of the brewing process, including brewer's spent grains, hot trub, spent hops, and brewer's spent yeast, are often conducted to animal feed or discarded straight into the soil for composting [5].

These brewing by-products present high nutritional values concerning macro- and micro-components, such as proteins, polysaccharides, b-glucans, vitamins, minerals, polyphenols, essential oils, and flavor substances, becoming an attractive raw material for food application [6]. Brewer's spent grains are the brewing by-product most used in foodstuff and food processes, followed by brewer's spent yeast due to its high protein content and fibers [7,8]. In turn, hot trub and spent hops are underused and often discarded

in soil, which can become a sanitary issue since they are composed of substances with antimicrobial properties that inhibit the growth of some microorganisms [9].

In recent years, some review papers have encouraged the use of brewing by-products for several purposes, but the recovery of polyphenol and bitter acids has been less emphasized [6,10–15]. Silva et al. [11] reviewed literature about xanthohumol extraction from hops by-products, such as hot trub and spent hops, as well as the benefits of its consumption, showing the lack of studies related to this theme. In this context, we have searched for studies correlating the extraction of polyphenols and bitter acids from brewing by-products to summarize the knowledge gap and contribute to new biorefinery strategies. For this, we briefly contextualize the characteristics of brewing by-products, bitter acids, and polyphenols, indicating their potential uses in the food, cosmetic, and pharmaceutical sectors. Thus, we reviewed and discussed, through critical analysis, emerging technologies and alternative solvents that have been employed to extract bitter acids and polyphenols from brewing by-products. Additionally, we demonstrated some possibilities for isolation and concentration processes that could obtain purified compounds. Industrial aspects were also reviewed, considering patents, process scaling up, and the economic feasibility of brewing by-products to obtain new products.

## 2. General Aspects of Brewing By-Products Generation

Figure 1 shows the main by-products from the brewing process with the respective quantity generated. Beer production begins with grounded barley malt mixed with water, followed by a mashing step that aims to promote starch hydrolysis into reduced sugars. The wort is separated from grain solids during lauter tun, leaving behind the most representative brewing by-product (around 85%) called brewer's spent grains [16]. From that, the wort proceeds to boil since bitter acids need thermal energy to isomerize into more water-soluble compounds (iso-$\alpha$-acids) [17]. During wort boiling, malt protein denatures and interacts with polyphenols and bitter acids. Those compounds form colloidal particles that precipitate into a solid by-product, called the hot trub. Hot trub is generated at a low volume in the brewing process (around 2%), but it is one the most promising solid by-products for use in the phytochemical extraction process due to its unique composition of functionalized compounds. The hopped wort is fermented in the presence of yeast, converting sugar into ethanol. After fermentation/maturation, the precipitated yeast cells, called brewer's spent yeast, are removed to avoid off-flavor by cell lysates [18].

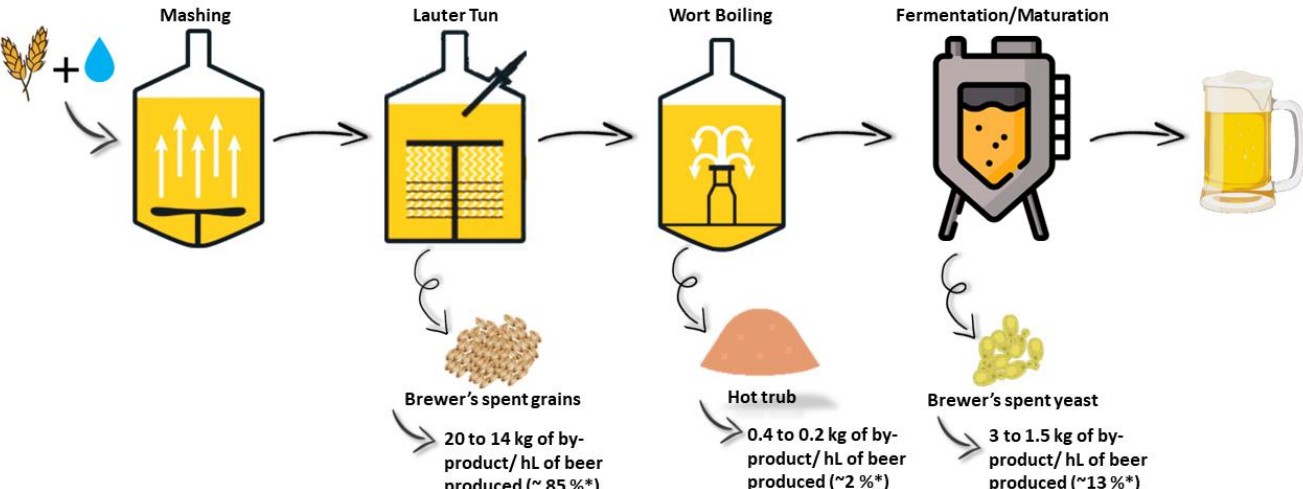

**Figure 1.** Scheme of beer manufacturing coupled with by-product generations per stage according to data from Mathias et al. [4]. * Percentage related to the volume total of by-products generated in all steps.

Considering the current market values of plant-based proteins, solid brewing by-products have potential for commercial use. In 2021 alone, more than 10.7 million protein tons were lost during beer manufacturing [3,8]. Beyond proteins, humulones (one of the bitter acids components) are high-priced compounds in the brewing industry. However, it is estimated that 50% of this compound is present in the hot trub, which accounts for around USD 8,368,740 per year in bitter acids losses by the brewing industry [4,19]. Therefore, developing alternatives for brewing by-products has both environmental and economic benefits.

### 2.1. Brewer's Spent Grains as a Source of New Products

The brewer's spent grains are obtained from the malt mashing step and contain high amounts of proteins, lipids, carbohydrates, fibers, minerals, phenolic compounds, and some vitamins, conferring a high nutritional value [6]. They are underexplored in the food industry as an ingredient due to the fast microbial growth that complicates the logistics chain. On the other side, several authors have explored the utilization of brewer's spent grains over the years, as seen in Figure 2, concerning the production of animal feed, protein extraction, and ingredient formulation, as examples.

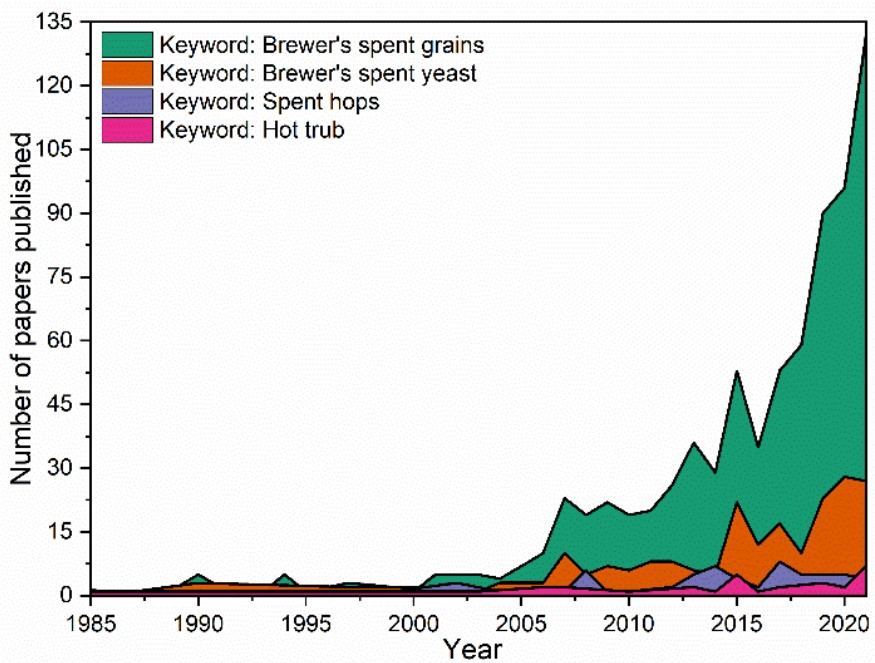

**Figure 2.** Data acquisition from Scopus databases.

Protein extraction is the primary use approach for brewer's spent grains, and precipitation at the isoelectric point is the most applied technique. Several protein solubilization methods are applied to increase yields in this technique, such as temperature variation and emerging technologies associated with using organic solvents, salts, and detergents, as can see in Table 1. At the beginning of protein recovery studies from food by-products, Ervin et al. [20] applied the temperature variation, which tests at 100 °C and resulted in a significant increase in protein content (~29% more nitrogen) compared with lower temperatures. Recently, emerging technologies have studied how to increase the extraction yield and promote modification in proteins. Another use of brewer's spent grains is the recovery of lignin, hemicellulose, and xylooligosaccharides for producing films and biofuels [21–24]. In this way, Moreirinha et al. [25] developed films composed of arabinoxylan nanofibers and cellulose from brewer's spent grains, proposing the production of food packaging with biodegradable characteristics.

**Table 1.** Studies about used compounds from brewer's spent grains attractive to the food sector.

| Recovered Molecule | Product | Technique | Ref |
|---|---|---|---|
| Proteins | Proteins concentrate | Alkaline extraction associated with temperature variation (from 27 to 100 °C) and precipitation at the isoelectric point | [20] |
| Xylo-oligosaccharides | - | Hydrolysis via hydrothermal treatment followed by fractionation using anion-exchange chromatography and size exclusion chromatography | [21] |
| Lignin | - | pH adjustment to lignin precipitation | [22] |
| Ferulic acid and *p*-coumaric acid | - | Alkaline hydrolysis | [26] |
| Fiber | Snack product | Extruded chickpea-based using brewer's spent grains as replacement maize flour | [27] |
| - | Baked snacks | Snacks formulation using brewer's spent grains as replacement wheat flour | [28] |
| Proteins/hemicellulose | Proteins and arabinoxylans | Sequential extraction of proteins and arabinoxylans using an alkaline solution and ethanol | [29] |
| Xylan | - | Alkaline extraction followed by ultrafiltration and diafiltration | [30] |
| Hemicellulose | - | hydrolysis via flow-through subcritical water reactor | [23] |
| Proteins/fibers | Proteins and fiber concentrate | Fractionation process via the combination of wet milling with chemical and enzymatic incubation | [31] |
| Arabinoxylans | Films | Arabinoxylans extraction via precipitation in pH adjustment followed by microwave treatment. The film was produced by the casting method. | [25] |
| Ferulic acid | - | Extraction via dilute acid hydrolysis followed by alkaline hydrolysis | [32] |
| Proteins/fibers | Proteins and fiber concentrate | Fraction process via wet milling with enzymatic incubation | [33] |
| Proteins | Fishmeal | Extruded | [34] |
| Fiber/proteins | Pork sausages | Brewer's spent grains flour as the replacement of a small part of pig meat | [35] |
| Fiber/proteins | Hybrid sausages | Brewer's spent grains flour as the replacement for broccoli and insect flour | [36] |
| Fiber/proteins | Dry Pasta | Extrusion | [37] |
| Xylose/arabinose/Amino acids | - | Hydrolysis via flow-through subcritical water reactor | [24] |

Moreover, phenolic compounds [26,32] and fibers [27,28,35,36] have been recovered from brewer's spent grains to obtain food and pharmaceutical ingredients. Campos et al. [35] and Talens et al. [36] replaced part of a meat product's formulation with brewer's spent grains flour to increase the fiber and protein content. The addition did not alter the texture of the products compared with the control. In this way, the brewer's spent grains have been widely used in baked [38] snacks [27,28] and pasta formulations [37] as fiber supplements (or as functional ingredients since they help in the gastrointestinal tract). These use options from brewer's spent grains show the high potential for use in the food industry, which will allow the growth of sustainability with the decreased cost of the matter source.

### 2.2. Hot Trub and Spent Hops as a Source of New Products

The brewing by-product generated after boiling hopped wort, characterized by its particulate nature, is referred to by various names in the literature. Among these, spent hops is the most-used term. However, the spent hops term also is used to describe a by-product of the extraction industry, which produces hops products, including bitter acids and essential oil extracts, via supercritical technology with carbon dioxide or reflux technologies using hexane (the oldest version [39]). The lack of standardized terminology has resulted in confusion in the literature regarding the intrinsic characteristics of each

by-product, especially for those new to this field. To address this issue, this review proposes a standardized nomenclature for by-products based on their composition. By-products containing malt proteins, hops compounds, and spent leaves are classified as "hot trub", whereas those composed solely of bines and spent leaves are classified as "spent hops". This proposed terminology will aid in the clarification and consistency of future research in this area.

As described before, hot trub is formed during wort boiling step in which malt proteins denature by long exposure to high temperatures (approx. 100 °C to 60 min, depending on the protocol of each brewery). The protein denaturation enables the crosslinking between proteins (protein–protein), bitter acids (protein–bitter substances), and polyphenols (protein–polyphenol) due to exposure of the hydrophobic sites that lead to aggregate formation and, consequently, precipitation [40,41]. This precipitation is required to clarify the brewer wort, avoiding colloidal instability in further steps as well as in packed beer. However, the precipitation also leads to undesired losses of an expressive quantity of bitter acids and polyphenols, influenced by the compound's solubility and the binding with protein aggregates. Those compounds are removed with the trub in the whirlpool. During this step, the material's deposition is promoted by centrifugal force in the equipment's center [18].

The potential uses of hot trub are varied due to its protein and bioactive compound contents. However, studies about the recovery of compounds' fractions are still limited. In the literature, the main recovery process involves emerging technologies as well as solvent extraction. Table 2 presents some studies evaluating the use of hot trub as a source for food processing.

**Table 2.** Studies about compounds used from hot trub and spent hops that are attractive to the food sector.

| By-Products | Molecule Recovered | Products | Techniques | Phytochemical Yields | Ref |
|---|---|---|---|---|---|
| spent hops | Desmethylxanthohumol/ xanthohumol | - | Extraction by polarity using apolar solvents | Xanthohumol (0.04%) Desmethylxanthohumol (0.00056%) | [42] |
| spent hops | Essential oil | - | Extraction via hydro-distillation | β-Myrcene (241 µg/g) Limonene (7.83 µg/g) Linalool (8.08 µg/g) Geraniol (3.08 µg/g) 2-undecanone (1.36 µg/g) β-caryophyllene (11.2 µg/g) α-humulene (19.1 µg/g) | [43] |
| spent hops | Bitter acids | - | Extraction by polarity using apolar solvents | Spent hops before isomerization: Isocohumulone (65.59 µg/g) Isohumulone (134.97 µg/g) Isoadhumulone (21.09 µg/g) Cohumulone (175.91 µg/g) N+adhumulone (580.70 µg/g) Colupulone (57.42 µg/g) Adlupulone (15.65 µg/g) Spent hops after isomerization: Isocohumulone (129.76 µg/g) Isohumulone (399.78 µg/g) Isoadhumulone (51.9 µg/g) Cohumulone (44.39 µg/g) N+adhumulone (58.73 µg/g) Colupulone (21.41 µg/g) Adlupulone (10.68 µg/g) | [44] |
| spent hops | Xanthohumol/ isoxanthohumol/6-prenylnaringenin/8-prenylnaringenin | Encapsulated in green gelatin capsules | Extract was produced via the enzymatic approach | 8-Prenylnaringenin (0.42 g/100 g dw) 6-Prenylnaringenin (2.18 g/100 g dw) Isoxanthohumol (1.35 g/100 g dw) Xanthohumol (35.78 g/100 g dw) | [45] |
| spent hops | Xanthohumol/ Isoxanthohumol | - | Extraction via Supercritical carbon dioxide | - | [46] |

**Table 2.** *Cont.*

| By-Products | Molecule Recovered | Products | Techniques | Phytochemical Yields | Ref |
|---|---|---|---|---|---|
| spent hops | Proteins/ Xanthohumol | - | Extraction using deep eutectic solvents | Xanthohumol (from 0.52 to 1.92 mg/g SH) Protein content (40 and 64%) | [47] |
| spent hops | Xanthohumol | - | Extraction via deep eutectic solvents | Xanthohumol (2.30 mg/g SH) | [48] |
| spent hops | Xanthohumol | - | Extraction by polarity using apolar solvents was followed by the bioconversion method (via enzymes) to 8-prenylnaringenin | Xanthohumol (from 0.36 to 2.14 %m) | [49] |
| spent hops | Polyphenols | - | Extraction by polarity using water and ethanol | α-Acids (0.011%m) Total polyphenols (10.45 g/L) Humulinones (0.031%m) | [50] |
| hot trub | Essential oils | - | Extraction via hydro-distillation | Myrcene (24.2 %) a-Humulene (16.2%) b-Caryophyllene (6.6%) 2-Undecanone (4.7%) Humulene oxide II (4.0%) 2-Methylbutyl isobutyrate (3.6%) d-Cadinene (3.3%) Linalool (1.9%) Limonene (1.2%) | [51] |
| hot trub | Proteins | Ice cream | Proteins flour of hot trub was added in the ice cream formulation as proteins enrichment | - | [52] |
| hot trub | Phenolic compounds | - | Extraction by polarity using water, methanol, and acetone | TPC (24.84 μmol GAE/g w.s.) | [53] |
| hot trub | Phenolic compounds | - | Extraction by polarity using water and hydroethanolic solution (70% *v/v*) | TPC (from 7.40 to 15.98 μmol GAE/g) | [54] |
| hot trub | Proteins | Proteins isolated | Alkaline extraction and precipitation at the isoelectric point | Protein content (94.56 %dw) TPC (35.36 mg GAE/g) Flavonoids (4.06 mg QE/g) | [55] |
| hot trub | Proteins | Pasta | Proteins flour of debittered hot trub for pasta production | Protein content (45.72%) | [56] |
| hot trub | Bitter acids | - | Extraction by polarity using hydroethanolic solution (30% *v/v*) and pH 7 that was followed by membrane filtration | N+-adlupulone (6.4 mg/g) Colupulone (6.3 mg/g) Iso-α-acids (28.23 IBU) N+-adhumulone (2.3 mg/g) Cohumulone (2.1 mg/g) | [57] |

dw—dry weight; SH—spent hops; %m mass percentage; w.s.—wet sample; TPC—total phenolic content; GAE— gallic acid equivalent; QE—quercetin equivalent; IBU—international bitterness unit scale.

Recently, a study by Silva et al. [57] proposed a recovery of bitter acids from hot trub using membrane technology, which is a promising alternative for producing bitter acids. High yields of phenolic compound recuperation were obtained using ultrasound-assisted technology under an ethanolic solution [58]. This fraction, comprising a high-added value hops polyphenol named xanthohumol, is currently of major interest to the pharmaceutical industry. Furthermore, hot trub has a high protein content, providing a low-cost source with plant-based characteristics [4]. A study by Saraiva et al. [55] proposed a simple extraction process using water under alkaline conditions to effectively recover the proteinaceous fraction from hot trub. This creates the possibility for an economically feasible use of plant-based proteins, which is a trend in the food industry.

A crucial part of the use of agro-food by-products is their application to add value to the by-product and, consequently, diminish their environmental impact. Over the past decades, hot trub has been combined with brewer's spent grains and used mainly for livestock feed. However, hot trub proves great application potential not only in food but also in the pharmaceutical industry. For example, Saraiva et al. [52,56] explored the application of hot trub isolated protein in food matrices, such as pasta and ice cream, enhancing the products' quality and acceptability. Hot trub bioactive compounds showed promising usage as a repellent against stored-food pests in replacement to synthetic ones [51]. Additionally, Censi et al. [54] indicates hot trub as a source of antioxidant phenolic compounds (quantified by TPC, FRAP, DPPH cell culture, cytotoxicity, and mitochondrial activity) with enormous potential to be applied in cosmetics formulations. The spent hops are a vegetal by-product from producing secondary hops metabolite extracts. The main approach to reusing spent hops is the extraction of residual phytochemical compounds, such as essential oils, phenolic compounds, and bitter acids (Table 2). The protein content was proposed as a use of spent hops by Wallen and Marshall [39]. However, over the years, this use has lost interest because other beer by-products have higher protein contents. Oosterveld et al. [59] studied pectin extraction to increase the valorization of spent hops, but it has a low production yield (~2%), becoming economically unfeasible. Phenolic compound recovery is the most exciting use of spent hops, from an economic point of view, due to it being a cheap raw material with a unique composition profile. Some polyphenols are only found in hops, for example: xanthohumol, dihydroxanthohumol, 8-prenylnaringenin, and desmethylxanthohumol, which are undergoing seasonality and high prices.

Chadwick et al. [42] performed an extraction via polarity using methanol, followed by an extensive molecule characterization of polyphenols from spent hops, aiming at producing an estrogenic extract. They concluded that 8-prenylnaringenin was the most powerful estrogen molecule formed via the chemical isomerization of desmethylxanthohumol. Moens et al. [49] produced 8-prenylnaringenin via bioconversion in an enzymatic route from xanthohumol extracted from spent hops. Jerkovic and Collin [60] used spent hops to produce resveratrol-enriched extracts, reaching 7.7 and 0.7 mg/kg trans-piceid and trans-resveratrol via ethanolic solution (50% *v/v*), respectively. Conventional strategies were also applied to extract phenolic compounds, as reported by Anioł et al. [43], in the recovery of essential oils from spent hops using water as a solvent in the distillation technique, and Anioł and Żołnierczyk [44] employed several organic solvents via reflux and ultrasonic device for bitter acids recovery. Nowadays, emerging technologies and green solvents (such as deep eutectic solvents and subcritical water) have been studied for phytochemicals from spent hops and hot trub, which will be discussed in the next section.

### 2.3. Brewer's Spent Yeast as a Source of New Products

The brewer's spent yeast generated after the fermentation step of hopped wort is the second brewing by-product, regarding to the volume of production, which corresponds to 13% of all by-products (Figure 1). This by-product is a rich source of proteins, carbohydrates, vitamins, and minerals, but its high level of nucleic acids limits its use as a human protein supplement due to the adverse health effects caused by excess uric acid [6]. During the last seven years, the use of brewer's spent yeast has been growing; Figure 2 shows the evolution of the publication numbers. These studies present varied content, from the benefits of consumption of brewer's spent yeast as animal feed to its use as a flavor enhancer food ingredient. Table 3 presents some examples of brewer's spent yeast as a source of attractive compounds for the food industry.

**Table 3.** Studies about compounds used from brewer's spent yeast attractive to the food sector.

| Recovered Molecule | Product | Technique | Ref |
|---|---|---|---|
| - | Meat substitute | Brewer's spent yeast hydrolysate was used as an ingredient in the formulation of meat products | [61] |
| Nucleotide | Flavor enhancer | Mechanic disruption treatment and chemical hydrolysis of RNA | [62] |
| β-glucan/mannoproteins | Mayonnaise stabilizing | Proteolysis treatment | [63] |
| - | Substrate for production of proteolytic enzymes | Brewer's spent yeast was used to support bacterial growth | [64] |
| β-glucan/proteins/proteolytic enzymes | Fortification of bread formulation | Chemical autolysis and mechanic disruption treatment | [65] |
| Phenolic compounds | - | Extraction by polarity using the hydroethanolic solution | [66] |
| Maillard conjugates | Wall material for encapsulated products | Enzymatic hydrolysis and heat treatment | [67] |
| Proteins/peptides | Emulsifying material and carrier agent for encapsulated products | Enzymatic hydrolysis was followed by ultrafiltration in concentration mode | [68] |
| Peptides | - | Enzymatic hydrolysis was followed by sequential membranes filtration | [69] |
| - | Emulsifying material and carrier agent for encapsulated products | The incorporation of carotenoids in brewer's spent yeast via ultra-turrax and ultrasonic methods was followed by atomization in spray dried | [70] |
| - | Wall material for encapsulated products | Encapsulation of curcumin into brewer's spent yeast using a pH-driven method and then lyophilized | [71] |

The primary use of brewer's spent yeast has been in producing proteins and polypeptide extracts beyond feed animals and, in uncommon circumstances, composting [6]. Higher added value applications have been evaluated using brewer's spent yeast in the food sector, such as ingredient formulation, protein replacement in meat and baked products, flavor enhancers, and food additives [7]. For any use purpose, the first step of the process is the rupture of the cell membrane of the yeast (via mechanical, chemical, or enzymatic approaches), followed by individual protocols of extractions. Silva Araújo et al. [63] promoted cell autolysis of brewer's spent yeast via the enzymatic route (proteolytic enzymes) and then followed that with mannoprotein and β-glucans extraction, which achieved yields of 40% and 10%, respectively. These mannoproteins were applied in a mayonnaise formulation that had its stability evaluated for 28 days, resulting in high emulsifying and stabilizing activity over the storage time, which would probably make it able to replace synthetic products (such as detergents) [63].

Aiming at the use expansion of brewer's spent yeast, the European Union (EU), in collaboration with Anheuser-Busch InBev (AB InBev), funded a project called Life Yeast (LIFE16 ENV/ES/000158) [72]. This project aimed at the development of a new methodology to process brewer's spent yeast as raw materials to obtain new products using technologies that are at or close to market readiness. During the period of project progress (from 2017 to 2019), 10% of all brewer's spent yeast from the EU was processed as a dry raw source to obtain animal feed, human supplementation, and other products [72]. Currently, the most valued use of brewer's spent yeast is as wall material for ingredient production, aiming at the entrapment of phytochemical compounds that could present some functionality to product conservation or even to the body's health. Several microencapsulation approaches have been applied to entrapping carotenoids, polyunsaturated fatty acids, vitamin C, and curcumin (Table 3). Marson et al. [67] used brewer's spent yeast as a wall material to encapsulate vitamin C via spray drying. They observed high entrapping efficiency with morphological characteristics typical of spray drying methods (a rounded and hollow shape with a semi-porous surface). Vélez-Erazo et al. [68] also produced particles via spray drying but rich in polyunsaturated oil. They observed a low

entrapping efficiency for the formulation made only with the brewer's spent yeast but with high protection against lipid oxidation. While Fu et al. [71] created microparticles via the pH-driven method for curcumin encapsulation, resulting in high entrapment efficiency and thermal stability with morphological characteristics, such as vesicles.

## 3. Characteristics of Bitter Acids and Xanthohumol

### 3.1. Bitter Acids

Bitter acids are from hops lupulin and are divided into two groups named α-acids and β-acids [73]. These fractions are composed of analog molecules that differentiate between them through the configuration of a side chain. The α-acids fraction is composed of cohumulone, humulone, adhumulone, prehumulone, and posthumulone, while the β-acids is composed of lupulone, colupulone, adlupulone, prelupulone, and postlupulone. From the α-acids, iso-α-acids (isohumulones and correlated substances) are generated through thermal isomerization during the wort boiling step [74]. The iso-α-acids are mainly responsible for the bitterness in beer due to higher solubility in aqueous solution [75]. Aside from that, researchers found an additional functionality of those compounds with regard to their antioxidant potential. According to Wietstock and Shellhammer [76] α-acids and iso-α-acid can deplete the formation of hydroxyl radicals due to metal chelation and scavenging ability. Oxidative reactions commonly occur during beer storage, reducing beer's quality. The antioxidant capacity of bitter acids improves beer's flavor stability during storage by avoiding oxidative changes and aldehydes formation [77,78]. Consequently, as these compounds degrade, the product loses its bitterness [79].

Antioxidant and antimicrobial properties of bitter acids offer different applications as food preservatives [80]. For instance, lupulone's addition to the chicken diet enhanced meat quality by protecting myofibrillar proteins from oxidation [81]. Another promising meat application was demonstrated by Kramer et al. [82], which verified the efficiency of bitter acids extract against *Listeria monocytogenes* in marinated meat. Apart from that, bitter acids extracts were efficient natural preservatives of sourdough bread [83]. While Xu et al. [84] applied the extract of bitter acids in film design based on gelatin and chitosan. Another study demonstrated the functionality of isolated β-acids as a preservative element in polylactic acid-based food packaging [85].

Few researchers reported the bitter acids' effects on human health until the first decade of the 2000s. Recently, a considerable amount of literature has been published on bitter acids' properties related to anti-inflammatory, sedative, antiacne, anticancer, and antioxidant effects. A series of studies conducted by Ano et al. [86–89] attempted to prove the anti-inflammatory role of isohumulones on the brain, improving memory and cognitive function as well as preventing neurological disorders, such as Alzheimer's disease, depression, and dementia [86–89]. The healthful properties of bitter acids are beyond giving beer flavor. Vast potential for application in cosmetics and natural medicines has drawn enormous attention from the pharmaceutical industry. However, further studies on those bioactive compounds' applications, toxicology, and posology need to be developed to create new products.

### 3.2. Polyphenols

Phenolic compounds, also known as polyphenols, are a diverse group of naturally occurring compounds found in plants. They are characterized by one or more aromatic rings with hydroxyl radicals and are produced by secondary metabolites throughout the development of plant tissues. In stressful situations, these compounds act as antibiotics, natural pesticides, and protective agents for plants. Polyphenols have a wide range of molecular structures, from simple aromatic chains (phenolic acids) to highly polymerized substances (tannins) [90]. The commercial interest in these compounds is due to their potential health benefits, including anti-inflammatory, antioxidant, and anticarcinogenic properties when consumed regularly. Additionally, polyphenols contribute to the sensory properties of food and beverages of plant origin, such as color and astringent flavor [91,92].

Malt and hops are the main sources of polyphenols found in beer and, consequently, in brewing by-products. In malt, compounds such as proanthocyanidins, chlorogenic, quinic, caffeic, sinapic, gallic, syringic, ferulic, and *p*-coumaric acids can be found [93,94]. Concerning hops flowers, polyphenols content reaches about 4–14% of dry weight, with phenolic acids, chalcones, flavonoids, and proanthocyanidins being the main components [95]. These polyphenols present bioactivity relevant for commercial purpose, e.g., the use of 8-prenylnaringenin as an estrogen precursor to relieve symptoms of menopause, as well as catechin and epicatechin to inhibit the growth of telomerase in human prostate and breast cancer cells [96]. Among the polyphenols in hops, proanthocyanidin and xanthohumol (prenylated flavonoid) are the most researched due to their reactivity with macromolecules, antioxidant capacity, and anticancer action.

Xanthohumol is especially notable among hops polyphenols due to its anti-inflammatory action against cancer cells, sterol regulatory ability, and also the capacity to prevent liver and heart disease [11]. It is only found in hops flowers, which represent 0.1 to 1% of their dry weight, and is secreted by lupulin glands (Figure 3) as a hard resin [97]. Due to its low solubility in water, xanthohumol does not significantly contribute to beer flavor (1.99 to 81.22 μg/L of xanthohumol in beers [98]). During the brewing process, the xanthohumol is thermally isomerized to iso-xanthohumol that is more soluble in water [99]. Non-isomerized molecules precipitate during the brewing wort boiling step and, consequently, are removed in the hot trub [100].

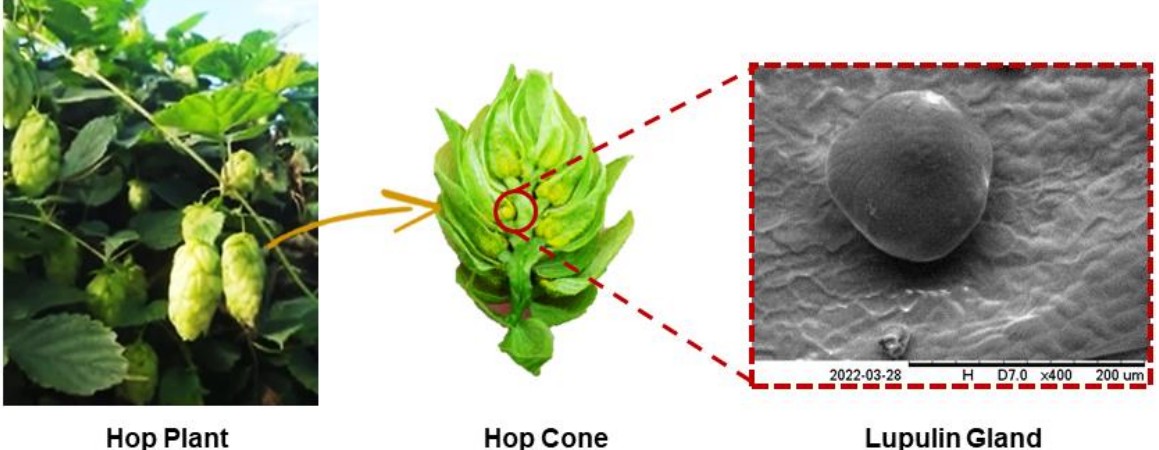

**Figure 3.** Hops flowers in situ, a transversal cut of hops cone, and lupulin glands via an image from an scanning electronic microscopic (data provided by authors).

The health human benefits of xanthohumol are still underestimated, with anti-cancer, estrogen precursors, antioxidation, anti-inflammation, and reduced rate of vascular fragility illnesses being the most widespread [97,99]. The anti-cancer effects of xanthohumol are the most studied in the literature via cell culture and animal experiments. Stevens [99] reported that xanthohumol could delay cancer cell proliferation through its anti-inflammatory property, which is induced by oxidative stress in the early stages. These effects are amplified when combined with chemotherapy drugs that do not allow resistance development against the cancer treatment. Silva et al. [11] reported that 1.35 to 2.5 mg. $kg^{-1}.day^{-1}$ is the minimum dosage in order for the human body to receive the pharmacologic effects of xanthohumol supplementation. However, an increase in studies on xanthohumol applications in beverages (beyond beers) and solids food is desirable for the comprehension of the interferences in flavor, toxicity degree, and bioavailability via simulation in the gastrointestinal system.

Another application of xanthohumol is in cosmetic products. The patent number WO-2010044076 claims the topical use of xanthohumol as the main, or even complementary, active ingredient in cosmetic formulations concerning products for brightening, anti-

redness agents, and anti-aging, as examples. Xanthohumol was efficient in decreasing skin cancer's effects and in improving skin elasticity in in vitro tests, as reported by Goenka and Simon [101]. In this sense, Philips et al. [102] studied xanthohumol as an active ingredient for replacing ascorbic acid (vitamin C) in skin aging products.

## 4. Trends for the Extracting Polyphenols and Bitter Acids from Brewing By-Products

The brewing by-products have been sources of different compounds for developing new ingredients, processes, and food products (Tables 1–3). Polyphenols and bitter acids stand out due to their high added value, as previously addressed in this review. Therefore, recent experimental studies have evaluated emerging technologies and deep eutectic solvents to obtain a higher extraction yield of these compounds. In this sense, we present, in Table 4, the most recent studies assessing the technologies of high-intensity ultrasound, microwave, high-pressure, ohmic heating, and pulsed electric field, exhibiting the process-optimized conditions and extraction yield by quantification for chromatography methods and/or total phenolic content (TPC). Furthermore, some examples of studies that evaluated alternative solvents to organic ones, such as deep eutectic solvents, are presented.

**Table 4.** Examples of the most recent studies assessing emerging technologies to obtain polyphenols and bitter acids from brewing by-products.

| Technology/Equipment | By-Product | Conditions Optimized | Results [1] | Ref |
|---|---|---|---|---|
| Ultrasound/1.2-cm probe system | brewer's spent grains | F (n-i), NP (49.5 W), T (25 °C), S/F (20), t (30 min), solvent (water) | TPC (0.66 mg GAE/g) | [103] |
| Ultrasound/13-mm probe system | brewer's spent grains | F (20 kHz), NP (750 W), pulse mode (5 sec on and 5 sec off), T (47 °C), S/F (21.7), t (30 min), solvent (20% ethanol aqueous solution) | TPC (3.55 mg GAE/g) *p*-hydroxybenzoic acid (10 μg/g) Ferulic acid (9.5 μg/g) Sinapic acid (13.5 μg/g) | [104] |
| Ultrasound/bath system | brewer's spent grains | F (37 kHz), NP (n-i), T (80 °C), S/F (20), t (50 min), solvent (65 ethanol: 35 water) | TPC (0.1 mg GAE/g) Ferulic acid (about 1.5 ± 0.2 mg/L), Vanillic acid (0.8 ± 0.2 mg/L) *p*-coumaric acid (0.12 ± 0.03 mg/L) | [105] |
| Ultrasound/bath system | hot trub | F (n-i), NP (n-i), T (n-i), S/F (4.46), t (60 min), solvent (ethanol) | TPC (8.93 mg GAE/g) TFC (1.31 mg QE/g) | [106] |
| Ultrasound and microwave/probe system and open-system microwave oven | brewer's spent grains | Ultrasound F (20 kHz), NP (45 W) Microwave F (2.45 GHz), NP (800 W) General information T (80 °C), S/F (30), solvent (72% ethanol aqueous solution), t (2 h) | Ultrasound TPC (4.11 mg GAE/g) Microwave TPC (3.91 mg GAE/g) | [107] |
| Microwave/domestic device | brewer's spent grains | Microwave pre-treatment F (2.45 GHz), NP (600 W and 800 W), t (30 min), sample mass (0.5 g) Magnetic stirring in an oil bath extraction 2 × [S/F (30), solvent (70 ethanol: 30 water, *v/v*), T (80 °C), t (60 min)] | Light barley: TPC (12.5 mg GAE/g) 4-Vinylguaiacol (2 μg/mL) Ferulic acid (34 μg/mL) *p*-Coumaric acid (7 μg/mL) Red barley: TPC (13.23 mg GAE/g) 4-Vinylguaiacol (0.7 μg/mL) Ferulic acid (14.83 μg/mL) *p*-Coumaric acid (9.1 μg/mL) | [108] |
| Microwave/system equipped 16-carrousel containers under magnetic stirring at 200 rpm | brewer's spent grains | F (n-i), NP (n-i), S/F (20), T (100 °C), t (15 min), solvent (0.75% NaOH aqueous solution) | TPC (13.7 mg GAE/g) TFC (2.6 mg QE/g) | [109] |
| Microwave/multiwave reactor with rotor-type equipment | brewer's spent grains | F (50 Hz), NP (n-i), S/F (10), t (13.3 min), T (100 °C), solvent [37.46% (*v/v*) water in the DES (1 choline chloride: 2 glycerol)] | TPC (2.89 mg GAE/g) 4-hydroxybenzoic acid (14.4%) Vanillic acid (7%) Vanillin (9.4%) Syringic acid (8.2%) Syringaldehyde (14%) Coumaric acid (20.6%) Ferulic acid (26.4%) | [110] |

**Table 4.** *Cont.*

| Technology/Equipment | By-Product | Conditions Optimized | Results [1] | Ref |
|---|---|---|---|---|
| High-pressure and ultrasound/pressurized liquid extractor and probe system | brewer's spent grains | Preheated conditions (1500 psi for 6 min), T (155 °C), t (17 min), P (n-i), solvent (35 ethanol: 65 water), S/F (11.33), Cycles (5) | TPC (17.2 mg GAE/g) | [111] |
| High-pressure/pressurized liquid extractor | rewer's spent grains | Preheated conditions (30 min at increasing pressure), P (5 MPa), T (185 °C), t (240 min), solvent (subcritical water), S/F (80) | TPC (33 mg GAE/g) | [112] |
| High-pressure and ultrasound/pressurized liquid extractor and probe system | malt rootlets | High-pressure Preheated conditions (1500 psi for 6 min), P (5 MPa), T (164 °C), t (15 min), solvent (ethanol:water mixture 33:67), S/F (n-i) Ultrasound F (n-i), NP (130 W), Amplitude (30%), S/F (50), T (room temperature), t (5 min) | High-pressure TPC (14.8 mg GAE/g) Ultrasound TPC (3.1 mg GAE/g) | [113] |
| High-pressure/supercritical $CO_2$ | brewer's spent grains | P (40 MPa), T (80 °C), solvent ($sCO_2$) | TPC (0.94 mg GAE/g) TF (0.219 mg QE/g) | [114] |
| Ohmic heating | brewer's spent grains | V (45 V to 75 V), distance between electrodes (8 cm), T (35 °C), t (30 min) solvents (ethanol: water mixture 60:40 $v/v$) | TPC (11.57 mg GAE/g) 4-hydroxybenzoic acid (125 µg/g) Vanillin (27.8 µg/g) Catechin (116 µg/g) Vanillic acid (23 µg/g) Ferulic acid (35.8 µg/g) | [115] |
| Pulsed electric field (pretreatment) | brewer's spent grains | Pretreatment conditions: E (2.5 kV/cm), F (50 kHz), t (14.5 s), S/F (1), solvent (water) | Free polyphenols (101 µg/g) Flavan-3-ols (10.1 µg/g) Flavonoids (60 µg/g) Phenolic acid derivates (55 µg/g) | [116] |

F: Frequency; n-i: non-informed; T: Temperature; S/F: Solvent/feed ratio; t: Extraction time; TPC: Total phenolic content; GAE: Gallic acid equivalent; TFC: Total flavonoid content; CE: Catechin equivalent; P: Pressure; V: voltage; E: electric field strength; [1] corelating conditions optimized.

### 4.1. High-Intensity Ultrasound Technology

High-intensity ultrasound (HIUS) has been enhancing the extraction of phytochemical compounds from different raw materials through acoustic cavitation [117]. This technology employs high power at low frequencies that promote compression and decompression of the liquid medium, causing cellular rupture and increasing mass transfer. Among the brewing by-products, the brewer's spent grains are the most applied for polyphenols extraction via HIUS technology. Only one study was found in which hot trub was use as a source of polyphenols and bitter acids extractions.

Zhang et al. [103] studied the extraction of polyphenols from brewer's spent grains using water by HIUS, observing that the highest TPC was recovered from raw materials with smaller particle sizes (100–250 µm) as compared to the larger ones (500–750 µm). The smaller particles provided higher surface area, increasing diffusion, mass transfer, and extraction yields. Based on the molecular weight and fragmentation pattern, they identified *p*-coumaric acid, caffeic acid, protocatechuic acid, catechin, and vanillic acid in the extract. Alonso-Riaño et al. [104] and Iadecola et al. [105] also demonstrated the efficiency of ultrasound over other extraction techniques (mechanical stirrer extraction and percolation) to obtain polyphenols from brewer's spent grains. These studies optimized the extraction conditions via quantifying TPC. However, Alonso-Riaño et al. [104] also quantified the individual compounds (*p*-hydroxybenzoic acid, vanillic acid, syringic acid, *p*-coumaric acid, vanillin, ferulic acid, and sinapic acid) in extracts produced by the best extraction conditions (Table 4) using water and hydroalcoholic mixture with 20% ethanol by chromatography. The ethanolic solution allowed a higher extraction yield of sinapic acid than water. According to the authors, water and solutions with low ethanol concentrations accessed the cells, but high ethanol concentration could cause protein denaturation, affecting the dissolution of polyphenols and influencing their recoveries. In addition, Alonso-Riaño et al. [104] also compared the ultrasound-assisted extractions with hydrolysis treatments (enzymatic, acid,

and basic) to obtain the polyphenols. In this second part of the study, they observed that the alkaline hydrolysis improved the extraction of all quantified polyphenols.

Senna Ferreira Costa et. al. [106] extracted polyphenols and bitter acids from hot trub using an ultrasound bath and ethanol as an extractor solvent. The ethanolic extract was dissolved in methanol–water solution (8:2 *v/v*) and exposed to sequential liquid–liquid partition to obtain the hexane fraction, ethyl acetate fraction, butanoic fraction, and aqueous fraction. Butanoic and ethyl acetate extracts presented higher TPC than the other extracts, demonstrating that most extracted compounds present medium or high polarity. However, the extracts presented low TPC values, which can be attributed to raw material composition acquired from an intense extraction process (step of wort boiling in the brewing industry), with only low polarity compounds remaining in the hot trub. Moreover, they identified the compounds extracted with each solvent via mass chromatography, observing that the butanolic extract presented cohumulone, cohulupone, hulupone/adhulupone, and the isoxanthohumol (resulting from xanthohumol isomerization); the ethyl acetate fraction presented cohulupone, hulupone/adhulupone, isoxanthohumol, and xanthohumol; and the hexane extract presented hydroxytricyclocolupone epimers, hydroxytricyclolupone/hydroxytricycloadlupone epimers, and nortricyclocolupone. According to the authors, cohulupone, hulupone/adhulupone, hydroxytricyclocolupone epimers, hydroxytricyclolupone/hydroxytricycloadlupone epimers, and nortricyclocolupone are probably obtained due to the severe oxidation processes of the β-acid components.

Therefore, high-intensity ultrasound technology has been increasing the recovery of polyphenols from brewer's spent grains. Only one study evaluated the extraction of compounds from hot trub using ultrasound technology, but they did not assess the effects of extraction variables. Instead, they used a predefined condition. As expected, the probe system has allowed better results of TPC than the bath system since the bath presents a high volume, and thus, less energy density is applied in the extraction medium. A similar result concerning a solvent–feed ratio (S/F) of approx. 20 shows the best relation between solvent quantity and raw material to favor the extractions. Water and hydroethanolic mixtures have been employed as solvents, following the principles of green chemistry. In addition, intermediate temperature (about 50 °C) and longer extraction times (>30 min) have allowed higher extraction yields. All the studies evaluated the extraction conditions quantifying the compounds by TPC. However, some identified and quantified the compounds in the extract acquired by the best parameters defined by TPC results. Among polyphenols, the most extracted from brewer's spent grains were *p*-hydroxybenzoic acid, ferulic acid, sinapic acid, vanillic acid, and *p*-coumaric acid. Cohumulone, cohulupone, hulupone/adhulupone, isoxanthohumol, xanthohumol, hydroxytricyclocolupone epimers, hydroxytricyclolupone/hydroxytricycloadlupone epimers, and nortricyclocolupone were identified in the extract acquired from hot trub. Thus, there is a gap of opportunity to study the effect of high-intensity ultrasound technology on the extraction of compounds other than brewer's spent grains from beer chains.

### 4.2. Microwave Technology

Microwave technology has been enhancing the extraction of polyphenols from different raw materials due to its heating effects on internal water molecules. In turn, these superheated molecules promote the cellular disruption of the cell walls of materials, facilitating the release of plant-bonded compounds [118]. Microwaves applied as pre-treatments or extraction techniques have been evaluated to extract polyphenols from brewer's spent grains (Table 3). Most of the studies compared microwave processes with thermal and ultrasound ones, demonstrating this technology's potential.

Zago et al. [108] evaluated a microwave pre-treatment on light and red barley brewer's spent grains to obtain higher yields of 4-vinylguaiacol, a bioactive compound derived from the thermal decarboxylation of ferulic acid. The pre-treatment time (from 5 to 30 min) at 600 W, which increased the temperature of the material from ~50 to ~100 °C, increased the extraction yield of polyphenols. The 4-vinylguaiacol content in the extracts increased

by about 80% when microwave pre-treatment was applied from 0 to 30 min, despite the power (600 and 800 W). However, the condition that allowed the highest TPC from red brewer's spent grains was 600 W for 30 min. They also used different solvents to perform thermal and high-intensity ultrasound extractions of the polyphenols from the brewer's spent grains. In this sense, they observed that the thermal treatment provided 40–60% (*w*/*w*) higher amounts of TPC than an ultrasound treatment due to the higher transfer mass allowed by increasing the medium temperature. Nevertheless, the comparison was unfair since they compared an ultrasound treatment of 1 min performed three times (totaling 3 min and an S/F of 30) with a thermal treatment performed for 60 min under stirring (300 rpm) at 80 °C at the same S/F of 30. Furthermore, the microwave pre-treatment followed by magnetic stirring in an oil bath extraction allowed higher extraction yields than both thermal and high-intensity ultrasound methods. Carciochi et al. [107] performed a coherent comparison between microwave and ultrasound technology to extract TPC from brewer's spent grains using solid-liquid extraction conditions previously optimized. They observed that these technologies increased TPC more than conventional solid-liquid extraction by 13% and 8% by ultrasound and microwave, respectively. Additionally, the extraction using ultrasound and microwave methods allowed higher extraction rates (2.42 and 1.66 mg/g/min, respectively) than conventional extraction (0.98 mg/g/min). According to the authors, ultrasound favored the extraction of polyphenols due to acoustic cavitation, as previously discussed in this review. On the other hand, microwaves increase the release of the compounds by combining heat and mass transfers working from the inside to the outside of the solid sample.

The solvent to assist the extraction performed by microwave technology has also been the subject of recent studies. Stefanello et al. [109] tested different solvents (50% methanol, 50% acetone, and 0.75% NaOH aqueous solution) to extract polyphenols from brewer's spent grains samples (defeated or integral ones), comparing microwave technology with a maceration procedure (performed using an S/F of 10 at ~20 °C for 24 h under constant stirring at 200 rpm). The aqueous solution of NaOH (0.75% *v*/*v*), the integral raw material, and the maceration extraction method were the most efficient conditions to obtain TPC. They assumed that the defeating process extracted some phytochemical compounds, mainly flavonoids, since the brewer's spent grains cell wall presents high lignin content. The microwave method could not promote sufficient molecular movement and rotation to overcome the barrier imposed by the phenolic compound's dimerization with the cell wall. Chromatographic analyses performed on the extracts obtained by maceration using 50% acetone and the alkaline solution demonstrated that the NaOH aqueous solution promoted structural simplification of polyphenols into monomers. Thus, the monomers derived from hydroxycinnamates (*p*-coumaric acid, trans-ferulic acid, and sinapic acid) were identified and quantified in alkaline extracts. However, acetone extraction quantified the native polymerized forms of polyphenols extracts. According to the authors, the alkaline solvent increased the antioxidant activity of the extracts, which could make this procedure interesting for pharmaceutical purposes. López-Linares et al. [110] observed the effects of a DES (1:2 choline chloride: glycerol) dilution in water and microwave temperature on the extraction of polyphenols from brewer's spent grains. Higher temperatures (from 50 to 100 °C) increased the polyphenols extraction, enhancing the solubility and diffusivity of the materials. Higher water concentrations in the DES acquired higher extraction yields at lower temperatures. On the other hand, the TPC decreased at higher temperatures when the amount of water in the DES increased. According to the authors, this behavior occurs because high water percentages in DES combined with high temperatures might break the hydrogen bonds between the DES components, which probably promotes the loss of the eutectic properties of the DES.

Different microwave devices presenting frequencies from 50 Hz to 2.5 GHz, applying powers of 600 and 800 W, have been employed to favor the extraction of polyphenols from brewing by-products (Table 3). Several studies did not present microwave device characteristics, making comparisons difficult. In general, treatments using microwaves

have been performed for shorter times than those using ultrasound (Table 3). However, in some cases, the longest treatment time could increase the extraction yields, as observed by Zago et al. [108]. In addition, the combination of alkaline solution and heat treatment helped increase TPC yields for polyphenols extraction from brewer's spent grains. Thus, microwave technology has only been evaluated to extract polyphenols from brewer's spent grains and could be further explored for other brewing by-products as raw material for compound recovery.

### 4.3. High-Pressure Fluid Technologies

High-pressure fluid technologies employ solvents at high pressures that present different physicochemical properties (viscosity, density, and dielectric constant) depending on pressure and temperature conditions, providing specific polarity and solvation power [119]. Among the different high-pressure techniques, pressurized liquid extraction (PLE) and supercritical carbon dioxide ($sCO_2$) have been evaluated to extract polyphenols from brewing by-products (Table 4). However, the studies that evaluated PLE and $sCO_2$ have focused on obtaining proteins or lipids from brewer's spent grains but also observed the co-extraction of polyphenols [111–114]. In this sense, considering that this part of the review aimed to observe the extraction of polyphenols and bitter acids, we focused on demonstrating the effects of high-pressure technologies on these constituents.

PLE employs different liquid solvents at high temperatures and pressures below their respective critical points [111]. These conditions favored solvent penetration in the sample matrix, decreasing solvent surface tension and viscosity and increasing mass transfer and the solubility of analytes, thus allowing higher extraction yields [120]. González-García et al. [111] evaluated PLE time (from 3 to 17 min), temperature (from 25 to 155 °C), and solvent percentage of ethanol (from 0 to 100%) on the extraction of polyphenols from brewer's spent grains. The intermediate rate of ethanol (35%) and the highest extraction temperature (155 °C) and time (17 min) allowed the highest TPC. After selecting these conditions, the authors evaluated the number of cycles (1–5), observing the highest extraction results after 5 cycles of extraction. Furthermore, comparing the results with other methods, they observed that PLE allowed better results than maceration using NaOH for 24 h (1.75 g GAE/100 g) and microwave extraction, also using NaOH, for about 1 h (1.38 g GAE/100 g). PLE extract that acquired the highest TPC also presented higher coumaric and trans-ferulic acid content. The presence of these polyphenols was also observed after gastrointestinal digestion, while p-hydroxybenzoic acid was identified only in PLE extracts and disappeared after gastrointestinal digestion.

An increase in temperature (from 125 to 185 °C) also increased the TPC value of extracts obtained from brewer's spent grains by PLE [112]. However, this increase was associated with newly formed compounds related to Maillard reactions since the extracts acquired at higher temperatures presented a visually browning color and higher amounts of hydroxymethyl furfural and furfural. In this sense, the authors tested hydroxymethyl and furfural standards, observing no answers concerning TPC analyses. On the other hand, they observed that products from a Maillard reaction increased the TPC assays in a text they performed. Additionally, different polyphenols were identified in the extracts acquired at 160 and 185 °C; both presented five polyphenols, two hydroxycinnamic acids, ferulic and *p*-coumaric acids, and three aldehydes: vanillin, protocatechuic aldehyde, and syringic aldehyde. Catechin, vanillic acid, 4-vinylphenol, and 4-vinylguaiacol were identified only in the extracted acquired at 160 °C. Otherwise, the extract obtained at 185 °C presented a higher content of aldehydes. Galván et al. [113], comparing high-intensity ultrasound and high-pressure technologies to extract proteins from malt rootlets, identified thirteen different polyphenols in the extract acquired by the PLE method. In contrast, none were identified in the ultrasound extract. The identified polyphenols were hydroxycinnamic acids, vanillactic acid, ethyl caffeate, sinapic acid, ferulic acid, and derived compounds.

As mentioned earlier, PLE and $sCO_2$ technologies strongly depend on the temperature and pressure conditions employed, which will regulate the solvent density and affect the

extracted components' yield, selectivity, and properties. Different solvents can be utilized above their critical point as supercritical solvents. However, $CO_2$ has been preferred since its critical point is at 31.1 °C and 73 atm. In this supercritical state, $CO_2$ presents high density, like a liquid, diffusivity, like a gas, and viscosity, like a gas–liquid [121]. The characteristics of $sCO_2$ have gained much attention as a solvent for extracting phytochemical compounds from different plant materials.

Alonso-Riaño et al. [114] evaluated the effects of brewer's spent grains particle sizes (1000 and 500 μm), temperature (40, 60, and 80 °C), and pressure (20, 30, and 40 MPa) on the extraction of polyphenols and flavonoids from brewer's spent grains using $sCO_2$. They observed that small particle sizes allowed higher solid extraction yields due to the higher superficial area they provided, which reduced the importance of diffusion concerning convection. Moreover, the highest values acquired for TPC and TFC were observed at the most elevated pressure (40 MPa) and temperature (80 °C) of $sCO_2$. The highest pressure allows higher solvating power associated with the higher $CO_2$ density. However, the extraction temperature presented the main effect on the extraction of phenolic and flavonoid compounds from brewer's spent grains. According to the authors, the temperature increase at high pressures ($\geq$15 MPa) raises the solute's vapor pressure. Thus, compensating for the decrease in $CO_2$ density with temperature increased the extraction yield. Furthermore, they discussed the low values of TPC and TF acquired for their extracts. They associated the low results with solvent polarity since some studies have reported that brewer's spent grains present more polar substances with low solubility in pure $sCO_2$. Thus, they suggest using ethanol as a co-solvent to increase the polarity of the extraction solvent.

Therefore, the high-pressure technology using water and hydroethanolic mixtures has allowed higher extraction yields of polyphenols from brewer's spent grains and malt rootlets. However, the $sCO_2$ technology did not favor the extraction of these compounds due to the solvent's polarity since polyphenols present in the brewing by-products present a more polar characteristic. High temperatures and pressures have favored the achievement of higher TPC values by PLE. However, these results have also been linked to Maillard reactions in the extractive medium. In this sense, it would be interesting to evaluate the profile of polyphenols obtained by all extraction conditions and not only in the best conditions determined by TPC. Thus, there is a gap of opportunity for further studies focused on extracting polyphenols from brewing by-products by evaluating PLE conditions and how they affect the profile of polyphenols in the samples.

### 4.4. Ohmic Heating Technology

The ohmic heating (OH) technology is based on applying electrical energy through the extraction medium dispersed in heat form due to the electrical resistance of its components, promoting electroporation and electrical breakdown of cellular structures, favoring mass transfer between raw material and solvent [122,123]. Bonifácio-Lopes et al. [115] performed the first unique study evaluating OH extraction to obtain polyphenols from brewing by-products. They tested two ratios of 60 and 80% ethanol: water (v/v) as solvent to obtain polyphenols from brewer's spent grains. The mixture using 60% ethanol allowed the highest TPC value and higher content and variety of polyphenols than 80%, demonstrating the compounds' affinity for a more polar solvent. The 60% ethanol solvent extracted 4-hydroxybenzoic acid, vanillin, catechin, vanillic acid, and ferulic acid, while the 80% did not extract vanillin and vanillic acid. Other studies have been reporting different compositions of polyphenols extracted from brewer's spent grains. However, this composition depends on the solvents, the S/F ratio, the temperature, the extraction methodology, the brewing methodology, and the raw material employed. The total amount of polyphenols quantified in extractions by chromatography was 39 times lower than the value determined by TPC. This difference can be attributed to the Folin–Ciocalteu reaction with other compounds. In alkaline conditions, Folin–Ciocalteu starts a reduction/oxidation reaction in the presence of, e.g., tryptophan, ascorbic acid, thiols, redox-active metal ions, and nucleotide bases substances, which produces a blue color that overestimates the polyphenol quantification [124].

Additionally, they observed that a conventional solid–liquid extraction (stirring for 30 min) allowed a higher TPC than the OH, but they did not evaluate the sample composition in polyphenols. Although conventional extraction provided higher yields, the ohmic heating technology still needs further evaluation since this study used only a pre-fixed condition.

### 4.5. Pulsed Electric Field Technology

Pulsed electric field (PEF) technology is based on the application of short pulses of electric voltage (from 0.5 to 20 kV/cm) in a medium placed between two electrodes [125]. This energy promotes a transmembrane potential that, after a critical voltage, generates repulsion between the charge-carrying molecules, forming pores in weak areas of the membrane and increasing the permeability [116]. PEF technology has been used as a pre-treatment for sequent extraction of phytochemical compounds from different raw materials [126,127].

We found only one study that evaluated PEF electric field strength E (0.5, 1.5, 2.5 kV/cm), frequency (50, 100, 150 Hz), and time of treatment (5, 10, 15 s) to obtain flavan-3-ols, flavonoids, phenolic acid derivates, and total free polyphenols from brewer's spent grains [116]. This study identified 13 free polyphenols: catechin-3-glucose, procyanidin B3, catechin, vanillic acid, quercetin-3-hexosylrutinoside, *p*-hydoxybenzaldehyde, vanillin, prodelphinidin B3, *p*-coumaric acid, hydroferuloyl glucose, ferulic acid, sinapoyl hexose, and tricin. After an extraction employing alkaline hydrolysis, they also identified and quantified the bond compounds in the extract acquired at optimized PEF conditions (2.5 kV/cm, 50 Hz, and 14.5 s). They were epicatechin, caffeic acid, *trans-p*-coumaric acid, *cis-p*-coumaric acid, trans ferulic acid, cis ferulic acid, sinapoyl-hexose (a, b, c, d, e, and f), and caffeoyl-hexose. The authors did not discuss the effects of PEF variables on extraction yields. However, they mentioned that other studies observed higher extraction yields by increasing the electric field strength from 1 to 3 kV/cm. Furthermore, the PEF pre-treatment at optimized conditions allowed 2.7 and 1.7 times higher free and bound polyphenols content than the extraction without the pre-treatment.

### 4.6. Deep Eutectic Solvents

Deep eutectic solvents (DES) are defined as binary or ternary mixtures of compounds that can associate mainly via hydrogen bond forming eutectic mixtures [128]. These eutectic mixtures are formed by complexing a hydrogen bond acceptor (HBA) with a hydrogen bond donor (HBD), presenting a melting point at a single temperature lower than the points of the separate constituents [129]. DESs have been used as alternative solvents to obtain xanthohumol and polyphenols from spent hops and brewer's spent grains [47,48,110].

Grudniewska and Pastyrczyk [47,48] evaluated choline chloride as an HBA and glycerol, ethylene glycol, propylene glycol, or lactic acid as an HBD mixed in a 1:2 molar ratio to extract xanthohumol from spent hops. In both studies, the first extraction tests were performed by a heated-stirring solid–liquid extraction procedure at 500 rpm and 60 °C for 1 h in an oil bath. The xanthohumol separation from DES was also carried out in the same way, adding water as an antisolvent to the DES. However, Grudniewska and Pastyrczyk [48] also evaluated the water content in DES (5, 10, 20, 30, and 40%), extraction S/F (10, 20, 40, and 50 *w/w*), the antisolvent to DES ratio (3:1, 4:1, and 5:1 *v/w*), extraction temperature (40, 50, 60, 70, and 80 °C), and extraction time (15, 30, 60, and 120 min). The highest xanthohumol extraction yield (2.30 mg/g) acquired by Grudniewska and Pastyrczyk [48] was obtained using the DES composed of choline chloride and propylene glycol with 5% of water (*w/v*) at S/F of 50 and 60 °C, using a ratio of 3:1 antisolvent/DES (*v/w*) for 1 h. However, acetone and methanol solvents allowed higher xanthohumol extraction of ~4.5 and 3.3 mg/g, respectively. Despite this, according to the authors, the traditional xanthohumol isolation and purification procedures using organic solvents are time-consuming and require many unsuitable steps for green chemistry applications. Moreover, the DES produced with choline chloride and propylene glycol could be used at least three times without decreasing the extraction yield. Grudniewska and Pastyrczyk [47] scale up the

process using higher amounts of raw material to perform the extraction. In this sense, the best composition of DES to extract xanthohumol from spent hops was choline chloride and lactic acid, achieving an extraction yield of 1.92 mg/g. The difference between the results was associated with the forming of precipitates that presented different particle sizes during the separation of xanthohumol from DES using water. According to the authors, using less raw material (0.5 g) allowed the formation of large aggregates, while more raw material (4 g) formed a precipitate with small particles. Therefore, the separation carried out through centrifugation at a large scale was insufficient to separate the obtained precipitates effectively. Thus, the author highlighted the antisolvent precipitation process as a critical step that must be individually selected for each DES.

Additionally, as previously reported, López-Linares et al. [110] studied different DESs and microwave conditions to extract polyphenols from brewer's spent grains. They evaluated choline chloride as HBA and ethylene glycol, lactic acid, glycerol, and 1,2-propanediol as an HBO at a molar ratio of 1:2 with the addition of 25% (*v/v*) in the DESs. They selected the best solvents using the same microwave-assisted extraction conditions at 65 °C for 20 min. The HBO that allowed the highest TPC (3.78 mg GA/g) was glycerol, which was different from that observed in the studies carried out by Grudniewska and Pastyrczyk [47,48] to extract xanthohumol from spent hops. Moreover, a higher extraction capacity of polyphenols was observed for polyalcohol than polycarboxylic acids (lactic acid). They also compared the extraction using a mixture of 80% methanol in water, observing lower results of TPC (1.2 mg GA/g) for the organic solvent.

Thus, using deep eutectic solvents as substitutes for organic solvents for extracting xanthohumol and polyphenols from brewing by-products has shown promising results. However, more efficient methodologies need to be developed to promote the separation of compounds from DESs. Furthermore, other HBAs can be tested for the formation of eutectic mixtures since only choline chloride has been studied so far.

*4.7. Comparison between Emerging Technologies and Alternative Solvents*

High-intensity ultrasound, followed by microwave and high-pressure, has been the most studied emerging technology for obtaining polyphenols and bitter acids from brewing by-products (Table 4). However, only one study evaluated the extraction of bitter acids from hot trub by ultrasound technology [106]. Moreover, this study did not assess the extraction possibilities by studying different parameters but used a pre-fixed condition (S/F of 4.5 at room temperature in a bath ultrasound system). Obtaining xanthohumol has only been studied using deep eutectic solvents, thus demonstrating the opportunity for future studies in the association of emerging technologies and DESs to acquire higher yields of xanthohumol and bitter acids from spent hops. Furthermore, solvents generally recognized as safe (GRAS) have been associated with emerging technologies to obtain polyphenols and bitter acids since most researchers used water and hydroethanolic mixtures to carry out the extractions.

Brewer's spent grains were the most studied to obtain polyphenols among the brewing by-products. This result was expected, considering the volume of brewer's spent grains generated during brewing process. Moreover, as previously discussed, hot trub, malt rootlets, and spent hops have been little studied, considering their content in phytochemical compounds. The polyphenols most frequently found in extracts obtained from brewer's spent grains are *p*-hydroxybenzoic acid, ferulic acid, sinapic acid, vanillic acid, *p*-coumaric acid, vanillin, syringic acid, and syringaldehyde. At the same time, xanthohumol was only recently studied in spent hops and hot trub.

The highest yields of TPC have been obtained by high-pressure technology (Table 4). However, as previously discussed, these can be unreliable since the reagent used for such a determination can react with other compounds in the medium, providing erroneous results. In this sense, most studies that evaluate the extraction conditions quantified the extracts by measuring the TPC. Some studies identify and quantify the polyphenols present in the extract that showed the highest TPC value. Thus, future studies should evaluate all

extraction conditions considering the profile of extracted compounds since a high TPC value does not always represent an increased extraction of polyphenols.

Based on the research reported in Table 4, Table 5 summarizes the main parameters used by each emerging technology.

**Table 5.** Summary of parameters used by each technology to obtain polyphenols and bitter acids from brewing by-products.

| Technology | Energy | Temperature | Extraction Time | S/F | Solvent |
|---|---|---|---|---|---|
| Ultrasound | F (20–37 kHz), Po (49.5–800 W) | 25–80 °C | 30–120 min | 4.46–20 | Water, ethanol, ethanol-water mixtures [103] |
| Microwave | F (50 Hz–2.45–GHz), Po (600–800 W) | 80–100 °C | 13.3–120 min | 10–30 | Ethanol–water mixtures, NaOH aqueous solution, solutions with DES [108] |
| High-pressure | P (5 MPa) | 155–185 °C | 15–240 min | 11.3–80 | Water, ethanol–water mixtures [111] |
| Ohmic heating | V (45–75 V) | 80 °C | 30 min | n-i | Ethanol–water mixture [115] |
| Pulsed electric field | E (2.5 kV/cm), F (50 kHz) | n-i | 14.5 s (pre-treatment) | 1 | Water |

F: Frequency; Po: Power; P: Pressure; V: Voltage; E: Electric field strength; n-i: non-informed.

The different forms of energy employed in each technology make comparisons difficult. However, ultrasound technology has been studied at different frequencies and powers to extract the phytochemical compounds from brewing by-products, even at low temperatures. This feature provides an advantage when the extracted compounds are thermolabile. Additionally, technologies that employ lower S/F are advantageous because they generate less waste in subsequent concentration and purification steps. Moreover, using nontoxic solvents for extractions is also a process advantage because it eliminates the necessity of subsequent purification steps. Therefore, ethanol–water mixtures and DES solutions present a better option for phytochemical extractions from by-products.

## 5. Methods of Isolation and Concentration of Bitter Acids and Polyphenols Obtained by Beer By-Products

The extraction approaches play a key role in separating molecules from the by-product matrices, but most techniques have low selectivity and isolation ability as a process limitation. The solvents applied presented affinity with a wide range of compounds during the bioactive substance extraction (such as polysaccharides, proteins, and soluble fibers), which need a complementary method with the proposal of isolation and concentration. Conventional methods such as coagulation, precipitation, adsorption process, and elution followed by solvent evaporation have been widely used in the hops industry. The present review investigated research that used nondestructive methods to purify bitter acids and xanthohumol components as adsorption and membranes technologies, aiming to minimize the compounds' deterioration during the process and show the different perspectives between the two techniques.

### 5.1. Adsorption Process

The adsorption process involves separating compounds from a fluid phase to a porous solid named adsorbent. The particles (adsorbate) are attached to the surface of the adsorbent by physical or chemical interactions during the selective process [130]. In physical adsorption, the adsorbate molecules are held to the surface through van der Waals forces that are relatively weak. In contrast, in chemical adsorption, the molecules are bound to the surface through chemical bonds, including electrostatic forces and hydrogen bonding. After the adsorption process, the molecules can be recovered from the adsorbent using fluids of different polarities that weaken the initial interactions [130,131]. The non-destructive character of the adsorption process has attracted considerable attention in the purification and separation industries due to its simplicity and eco-friendly nature. However, the

adsorption process has limitations, as adsorbents' expensive cost, regeneration capacity, and recycling techniques discourage large-scale reproduction [132].

Adsorption techniques have been used on large production scales to separate xanthohumol from hops products for over 30 years [133]. Biendl [133] described a process that involves mixing hops and a 90% ethanol solution (*v/v*), followed by running the mixture through a continuous counter-current extractor. This procedure produces a raw extract containing polar hops compounds. The extract was then combined with diatomaceous earth and subjected to a sCO$_2$ process to separate bitter acids. The remaining residue is eluted using 96% ethanol as a desorption agent for polyphenol recovery, and its solvent evaporates to produce a yellow powder [133]. As a result of the isolation approach, the powders contain a high composition of xanthohumol beyond the content of other polyphenols (isoxanthohumol, 6-prenylnaringenin, and 8-prenylnaringenin). This study also reported the polyphenol concentration using polyvinylpolypyrrolidone resins in the adsorption process, achieving values from 9 to 80% xanthohumol. According to the author, this approach can be used in spent hops to obtain similar results in xanthohumol concentration. However, we did not find studies employing this separation technique to separate phenolic compounds from hops by-products, such as hot trub and spent hops, showing a knowledge gap.

In this way, Idea et al. [134] evaluated the partial purification of a ferulic acid extract acquired from brewer's spent grains via the adsorption process with synthetic resin (Lewatit VPOC1064 MD PH®). The authors optimized ferulic acid extraction by alkaline hydrolysis in the aqueous solution, where its concentration (by chromatography method) was monitored. The adoption kinetic was first performed with a ferulic acid standard, evaluating the effects of ferulic acid concentration, temperature, and resin proportion on the separation. Thus, they avoided other brewer's spent grains extract substances' interference, such as proteins, polysaccharides, and other polyphenols. After setting the adsorption condition, the partial purification of the raw extract was performed using ethanol solution at 70% (*v/v*) as an elution vehicle. This procedure resulted in the desorption of ~69% of ferulic acid. However, the authors did not present data about the composition of the final ferulic acid extract. Thus, this lack of data does not allow the interpretation of the purification results despite the nuclear magnetic resonance spectra ($^1$H NMR) presented by them, showing the disappearance of some peaks.

Solid-phase extraction and polyvinylpolypyrrolidone resins were commonly employed for clean-up, fractionation, and polyphenols concentration prior to chromatography analysis [135,136]. However, few studies have studied the adsorption methods to isolate and concentrate polyphenols from brewing by-products.

### 5.2. Membrane Technologies

Membrane technologies are based on the compounds' separation through semipermeable barriers, where fluids and solutes are transported and selected in the presence of transmembrane pressure ($\Delta$p) via a driving force. The semipermeable barriers are usually porous, which allows the transport of one or more chemical species. The pore size range of the membrane determines the compounds' rejection, which is mainly associated with the molecular weight cut-off (MWCO) and, with minor impact, the molecular structure, superficial charge, and hydrophobicity. Microfiltration is a membrane with the largest pores (from 100 to 10,000 nm) that separates suspended particles, allowing the permeation of macro and micromolecules, salts, and water. Ultrafiltration membranes have pore sizes between 2 and 100 nm, which can retain different macromolecules (such as suspended solids, carbohydrates, pectin, proteins, polypeptides, and phenolic compounds with high molecular weight) ranging from 350 to 1 kg/mol. Nanofiltration is characterized by pores from 2 to 0.5 nm that retain micromolecules with molecular weights from 1000 to 120 g/mol, such as anthocyanins, polyphenols, phenolic acids, sugars, and peptides. These filtrations are very well consolidated in several productive sectors due to the process advantages such as the nondestructive method, not involving phase changes or chemical agents, high selectivity, continuous process, simple operation, and scale-up ease.

In recent years, membrane technologies have been applied to add value to several agro-food by-products in the purification and concentration of bioactive compound extracts [137]. Polypeptides and proteins fractionation/concentrations are the typical applications of membranes technologies considering the brewing by-products, especially when it comes to brewer's spent grains and brewer's spent yeast. In contrast, spent hops and hot trub have fewer papers studying these processes, such as valorization protocols of low cost with eco-friendly natures to purify the compounds acquired from spent hops and hot trub. Mussatto et al. [138] proposed the use of brewer's spent grains via a biorefinery concept, where several products (xylitol, lactic acid, activated carbon, and polyphenols) were produced in different industrial steps in the same facility and where technical, economic, and environmental analyses were conducted. Regarding polyphenol recoveries, the authors extracted *p*-coumaric and ferulic acids from the black liquor stream provided by the soda pulping step followed by the sequential ultrafiltration and nanofiltration. They did not expose the final concentrations of polyphenol extract but simulated the diary productivity that reached 7.42 tons/day. This productivity was low comparing the other developed products. Nevertheless, polyphenol products are sold at high prices in low quantities, which makes this process economically feasible [138].

Silva et al. [57] reported a sequential membrane process in a crossflow system (microfiltration followed by nanofiltration) to purify and concentrate the bitter acids extract from hot trub. This extract was produced by a solid–liquid approach with magnetic agitation, pH adjustment (pH 7), and ethanol solution at 30% (*v/v*). The authors add a diafiltration operation in microfiltration to improve the permeation of the bitter acids and decrease the fouling process. The fouling process is the main limitation of membrane technologies, which reduces their flux, selectivity, and process time because it requires interruption of the operation to promote cleanup. The fouling issue also was a process limitation of Varga and Márki [139] during the hot trub removal from hopped wort via microfiltration in the crossflow system. Despite that, Silva et al. [57] obtained a rise in bitter acid permeation (from 69 to 92%) in the microfiltration addition of diafiltration, allowing the retention of suspended solids that could make the following process difficult. The nanofiltration step was successful, allowing an increase in the bitter acid content of almost three times without forming severe fouling.

Membrane technology has been shown to be promising to add value to polyphenols from brewing by-products, considering all that has been discussed above. However, more studies need to be performed to better understand the process variables (such as the variations in transmembrane pressure, crossflow velocity, and different module membranes). Furthermore, economic feasibility analyses must be performed to observe the financial aspects of using this technology. These technologies are well-established in several industrial sectors, which could encourage their application and funding.

## 6. Industrial Aspects: Value of the Extracted Compounds, Process Scale-Up and Economic Evaluation and Patents

The global trend of functional food consumption has expanded the polyphenol market, which achieved values of USD 1.6 billion just in 2021 and was projected to reach USD 2.9 billion by 2030 [140,141]. These data are correlated to the development of, e.g., food and functional beverages, teas, grape seeds, and dietary supplements where flavonoid compounds are the most utilized, followed by resveratrol and phenolic acids (such as ferulic acid). The consumption increase in these products is due to awareness of the benefits of maintaining good health. In this context, skincare is one of the health habits developed in recent years, and it represents a large consumption of polyphenols as active ingredients for antioxidant and anti-aging purposes. In 2020, the cosmetic segment was valued at USD 119 million and was projected to reach USD 158 million by 2025 [142]. The hops polyphenols are still underused for cosmetics, but we found one company in Poland, DermoTech Beauty, that has produced dermo-cosmetic products with xanthohumol as an active ingredient [143]. The most used hops phytochemical are the bitter acids, which

are irreplaceable due to their ability to give a bitter taste. The consumption of hops has been rising yearly since 2010 due to worldwide beer consumption, the search for better beer qualities, and the trends in highly hopped beer. In 2021, the hops market achieved production of 12.5 thousand tons of α-acids, an increase of close to 10% compared to 2019, as reported by Hopsteiner [144]. The market value of hops is based on the α-acid content, which explains the expansion of 13% of α-acid/ha over the past five years [145,146]. The price of α-acid accounts for around 143 EUR/kg, representing the overprice raw material in the brewing industry [147]. From all the points above, recovery polyphenols and bitter acids from brewing by-products are excellent opportunities, because of their low price and high availability. In addition, the recovery approach via by-products could decrease the stress on the fruit and vegetable chain and reduce the use of natural resources, producing a sustainable chain. Therefore, research and industries must work together to improve recovery approaches, increasing the economically viability of processes to acquire bioactive compounds from brewing by-products.

The studies of scale-up and economic viability are essential from a commercial point of view, demonstrating the feasibility of a recovery processes. Alonso-Riaño et al. [148] evaluated the feasibility of scaling up a process employing subcritical water at 170 °C for 22 min to obtain carbohydrates (monomer and oligomers), protein and free amino acids, polyphenols, and inhibitors, such as acetic acid, furfural, and HMF, from brewer's spent grains. They studied the process from a laboratory (0.5 L) to a pilot scale (25 L), observing good reproducibility of the scale-up in producing arabinoxylo-oligomers, gluco-oligomers, protein, and free amino acids. However, at the pilot scale, the yields acquired for xylo-oligomers were 13% lower than at the lab scale. Higher concentrations of monomers and polyphenols were also verified in the hydrolysates obtained at the lab scale. According to the authors, this difference is due to the preheating time. The heating rate that the system needed to reach 170 °C was 15.7 min on the lab scale, while it was achieved after only 4.5 min at the pilot plant. This heating time probably impacted the hydrolysis of some biopolymers.

The TPC acquired from brewer's spent grains before subcritical hydrolysis were 6.27 mg GAE/g and, after 45 min of isothermal treatment, increased to up 34 mg GAE/g. However, the pilot scale allowed 38% less TPC than the laboratory one, demonstrating that the higher preheating time at the lab scale allowed higher TPC. As mentioned previously, temperature plays an essential role in extracting polyphenols from brewing by-products, necessitating the consistent application of heat throughout all steps in the scaling-up process. Additionally, it is indispensable to note that quantifying TPC is not a reliable method for evaluating polyphenols from complex mixtures, as it may lead to overestimating results. Therefore, to obtain more accurate and reliable results, it is necessary to conduct chromatographic analyses to observe the profiles of extracted polyphenols.

We found only one study demonstrating the economic aspects related to the use of brewing by-products to obtain special flour. In this analysis, Colpo et al. [149] estimated the economic and financial feasibility of obtaining a special flour from brewer's spent grains by considering some parameters such as the minimum rate of attractiveness, the initial investment (working capital and equipment), monthly expenses, and an analysis time of 5 years in four scenarios: minimum production and minimum price, minimum production and maximum price, maximum production and minimum price, and maximum production and maximum price. The first scenario was unfavorable, showing a return on capital only after ten years of operation, but the others were favorable. After this simulation, authors performed others by the Monte Carlo method (random simulation). In this, they observed that, in 10,000 runs, there was no loss, the minimum profit was USD 2513, and the maximum was USD 42,542, determined according to Equation (1).

$$Profit = \{(supply \times price) - [(supply \times price) \times 20\%]\} - expenses \tag{1}$$

Another critical factor that was considered to determine production costs was the location of the factory, which must be close to industries that produce beer waste. Therefore,

in this study, they demonstrated the possibility of developing a new, innovative product by applying the concept of open innovation through the proposal of artisanal small and medium companies using raw materials that are by-products from other industries. More studies need to be carried out to analyze the scale-up and economic viability of processes employing brewing by-products.

Many patents have been developed associated with brewing by-products. A search in the Scopus database using the keyword "brewer by-products" found 1680 patents filed from 2012 to 2023. However, this number was limited to 908 when we searched for the keywords "brewer by-products" and "extraction". Many of the patents are focused on the brewing process, such as number WO2023285683, which proposes a system to monitor the quality of beer fermentation, and US20220380814, which provides a method and system for treating the microorganisms during propagation, conditioning, fermentation, and preservation. However, others have proposed biorefinery processes using beer by-products to obtain other products. The patents EP3307788 and US11130825 proposed pressurized hot water extraction procedures to extract hemicellulose from by-products. CN101138418B, developed in China, presented a comprehensive utilization of supercritical hops by-products as a source of an extract rich in phenolic compounds. A French group developed production methods of surface-active ingredients from spent brewers' grains.

## 7. Perspectives for Further Studies and Final Considerations

This review highlights the potential of brewery by-products as a valuable source of molecules of interest for the cosmetic, pharmaceutical, and food sectors. Among the brewery by-products, brewer's spent grains have been widely studied due to their high-volume production and composition, containing mainly proteins, fibers, and polyphenols in smaller proportions. Similarly, brewer's spent yeast has been utilized as a carrier agent for encapsulated products and as a source of proteins. Despite their high content of bitter acids and xanthohumol, spent hops and hot trub remain underutilized.

Among emerging technologies, ultrasound and high-pressure fluids are popular polyphenols and bitter acids extraction techniques. At the same time, microwave, ohmic heating, and pulsed electricity have been studied to a lesser extent. TPC quantification has been commonly used in optimization studies. However, this method lacks specificity for polyphenolics, which may lead to overestimating results in the presence of other substances, such as proteins and free sugars. Therefore, it is critical to employ analytical methods, including chromatographic assays, molecular profiling by nuclear magnetic resonance, and even individual identification by spectrophotometry to assess extraction efficiency.

Studies have shown that DES holds promise for recovering xanthohumol from spent hops, but more research is necessary to optimize the extraction process as well as in the use of other brewing by-products as raw materials. Therefore, there is a knowledge gap regarding brewing by-products, and future studies should focus on exploring additional aspects beyond extraction technologies. For instance, separation methods, such as adsorption and membrane technologies, need to be developed to allow their applications for producing the final products. Additionally, evaluating the economic feasibility and scalability of the process is essential, as patents related to the subject demonstrate a substantial industrial interest in reusing by-products from brewery industries.

**Author Contributions:** K.F.C.e.S.: writing—original draft, conceptualization, methodology, and investigation; M.M.S.: writing—review and editing, conceptualization, and investigation; M.B.C.P.: writing—review and editing and investigation; M.A.R.: writing—review and editing; M.D.H.: writing—review and editing, supervision, and funds acquisition. All authors have read and agreed to the published version of the manuscript.

**Funding:** This research was funded by FAPESP grant number 2019/27354-3; 2021/06863-7; 2021/12264-9, CAPES with Finance Code 001, and CNPq grant number 428644/2018-0; 309022/2021-5.

**Conflicts of Interest:** The authors declare no conflict of interest.

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
