# Peer review of "Processing Strategies for Extraction and Concentration of Bitter Acids and Polyphenols from Brewing By-Products: A Comprehensive Review"

_processes, doi:10.3390/pr11030921_

Round 1

Reviewer 1 Report

-Page2 In recent years, some review papers “please list some of these to provide the broad idea to readers”

-Page 7 Table3 “authors should add one more column about the yield for each study”

-Page 9 Additionally, the bioactivity “please list some examples of bioactive compounds”

-Page 11 Concerning hops flowers, polyphenols “please provide some examples of applications of these compounds or which biological activities”

-Page 11 high biological activity “please add detail of this”

-Page 11 Trends for the extracting polyphenols and bitter acids from brewing by-products. “authors should add another table to summarize pros and cons of each technology mentioned here”

-Page 20 Moreover, this study did not assess “what does it mean? please explain”

-Page 20 The highest yields of TPC “please add reference”

-Page 22 Industrial aspects: process scale-up, economic evaluation, and patents “authors should have some data of market, growth and product value”

Author Response

Page2 In recent years, some review papers “please list some of these to provide the broad idea to readers.”

Response: Thank you for the comment. We added some references.

Page 7 Table3 “authors should add one more column about the yield for each study.”

Response: Thank you for your suggestion. We added one more column with the yields.

Page 9 Additionally, the bioactivity “please list some examples of bioactive compounds.”

Response: Thank you for your suggestion. Indeed, adding the polyphenol components to our review could improve the available information for the reader. However, we cannot include this information because the cited study (Censi et al. (Censi et al., 2021)) quantified the phenolic compounds by TPC. We include this information in the manuscript.

Page 11 Concerning hops flowers, polyphenols “please provide some examples of applications of these compounds or which biological activities.”

Response: Thank you for the comment. We added some examples.

Page 11 high biological activity “please add detail of this.”

Response: Thank you for the comment. We added details.

Page 11 Trends for the extracting polyphenols and bitter acids from brewing by-products. “Authors should add another table to summarize pros and cons of each technology mentioned here.”

Response: Thank you for your suggestion. We presented another table summarizing pros and cons in the section 4.7.

Page 20 Moreover, this study did not assess “what does it mean? please explain.”

Response: Thank you for your comment. We reviewed the sentence.

Page 20 The highest yields of TPC “please add reference.”

Response: Thank you for your question. The references are the results presented in Table 4. We reviewed the sentence.

Page 22 Industrial aspects: process scale-up, economic evaluation, and patents “authors should have some data of market, growth and product value.”

Response: Thank you for your suggestion. We added some data of market, growth and product value in the section 6.

REFERENCES

Censi, R., Vargas Peregrina, D., Gigliobianco, M. R., Lupidi, G., Angeloni, C., Pruccoli, L., Tarozzi, A., & Di Martino, P. (2021). New Antioxidant Ingredients from Brewery By-Products for Cosmetic Formulations. Cosmetics, 8(4), 96. https://doi.org/10.3390/cosmetics8040096

Reviewer 2 Report

The document contains an extensive review of "Processing strategies for extraction and concentration of bitter acids and polyphenols from brewing by-products: a comprehensive review".

I think the work is of interest, given the global trend to use by-products from various industries, and the brewery is one of the most important worldwide.

However, to make the document more attractive to researchers, it is necessary to modify the abstract considering all the relevant points in the review. since it seems to me that although the figures are important, they can be part of the introduction.

Author Response

Response to Reviewer 2 Comments

The document contains an extensive review of "Processing strategies for extraction and concentration of bitter acids and polyphenols from brewing by-products: a comprehensive review."

I think the work is of interest, given the global trend to use by-products from various industries, and the brewery is one of the most important worldwide.

However, to make the document more attractive to researchers, it is necessary to modify the abstract considering all the relevant points in the review. Since it seems to me that although the figures are essential, they can be part of the introduction.

Response: Thank you for the comment. We made some alterations to the abstract, as suggested to you.

Reviewer 3 Report

The manuscript „Processing strategies for extraction and concentration of bitter acids and polyphenols from brewing by-products: a comprehensive review” was submitted to Processes for publication.

Broad comments:

The review article gives a good overview on the possibilities for using by-products from industrial production and also show the current methodologies to yield such compounds, in this specific case from the beer brewing process.

I have two comments on the structure of the article and several minor points.

The first comment is on the format of the tables. I would suggest to move the first column (references) to the back. Furthermore, citing the names in the tables can be avoided and giving only the number of references is sufficient. Thus, more space is generated to eventually broaden the column with, i.e., the used techniques. This accounts for all tables in the manuscript.

Moreover, in table 4 a sorting of lines should be conducted by either the by-product or the used technology (or one after the other). If doing the second way, then columns 2 and 3 would have to be switched.

The second structural comment is on section 2. Here, I would suggest to create 3 subsections, namely 2.1 spent grains, 2.2 hot trub/spent hops, 2.3 spent yeast. This is more chronological. It should also be made clear that the spent hops comes along with the hot trub. Maybe by inserting this into figure 1. Also, in table 3, dividing hot trub and spent hops is not ideal as it suggests that it is two different things. At the same time bitter acids are described to come from the hot trub and from the spent hops. Of course, there are techniques, e.g., dry hopping, but I think this was not meant, but if so, then it should be explained before.

Additionally, the last paragraphs of section 3.1. and 3.2. are two specific and should be re-written in a more general way or deleted.

Minor comments:

Line 6 in the introduction:           Please write “Mathias et al.” instead of “Mathias, Mello, and Sérvulo”

Immediately after Figure 2:         Please write “Protein extraction” instead of “Proteins extraction”.

General:                                              Please write the “p-“ in e.g., p-coumaric or p-ferulic acid in italics.

Some sentences need improvement:     E.g., The first sentence after table 1 or the first sentence in section 3.1.

Supercritical carbon dioxide:      The authors use sCO2 and sc-CO2 but only explain the first abbreviation. I guess the second should mean the same?

3.6 Deep eutectic solvents should be 4.6 deep eutectic solvents

In the second paragraph of this section please write “Grudniewska and Pastyrczyk [57,65]”

Author Response

Response to Reviewer 3 Comments

The manuscript “Processing strategies for extraction and concentration of bitter acids and polyphenols from brewing by-products: a comprehensive review” was submitted to Processes for publication. 

Broad comments:

The review article gives a good overview on the possibilities for using by-products from industrial production and show the current methodologies to yield such compounds, in this specific case from the beer brewing process.

I have two comments on the structure of the article and several minor points.

The first comment is on the format of the tables. I would suggest moving the first column (references) to the back. Furthermore, citing the names in the tables can be avoided and giving only the number of references is sufficient. Thus, more space is generated to eventually broaden the column with, i.e., the used techniques. This accounts for all tables in the manuscript.

Response: Thank you for your suggestion. We reviewed the tables according to your comment.

Moreover, in table 4 a sorting of lines should be conducted by either the by-product or the used technology (or one after the other). If doing the second way, then columns 2 and 3 would have to be switched.

Response: Thank you for your suggestion. We reviewed the table according to your comment.

The second structural comment is on section 2. Here, I would suggest creating 3 subsections, namely 2.1 spent grains, 2.2 hot trub/spent hops, 2.3 spent yeast. This is more chronological. It should also be made clear that the spent hops come along with the hot trub. Maybe by inserting this into figure 1. Also, in table 3, dividing hot trub and spent hops is not ideal as it suggests that it is two different things. At the same time bitter acids are described to come from the hot trub and from the spent hops. Of course, there are techniques, e.g., dry hopping, but I think this was not meant, but if so, then it should be explained before.

Response: Thank you for your suggestion. We kept our idea of differentiating between the two by-products and changed the position of the paragraph. Indeed, this change could improve the understanding of the reader.

Additionally, the last paragraphs of section 3.1. and 3.2. are two specific and should be re-written in a more general way or deleted.

Response: Thank you for your suggestion. We reviewed the manuscript according to your comment. 

Minor comments:

Line 6 in the introduction:           Please write “Mathias et al.” instead of “Mathias, Mello, and Sérvulo”.

Response: Thank you for your comment. We reviewed the manuscript according to your suggestion. Immediately after Figure 2: Please write “Protein extraction” instead of “Proteins extraction”.

Response: Thank you for your comment. We reviewed the manuscript according to your suggestion. General: Please write the “p- “in e.g., p-coumaric or p-ferulic acid in italics.

Response: Thank you for your comment. We reviewed the manuscript. Some sentences need improvement: E.g., The first sentence after table 1 or the first sentence in section 3.1.

Response: Thank you for your comment. We reviewed the manuscript according to your suggestion.

Supercritical carbon dioxide:      The authors use sCO2 and sc-CO2 but only explain the first abbreviation. I guess the second should mean the same.

Response: Thank you for your comment. We reviewed the manuscript according to your suggestion.3.6 Deep eutectic solvents should be 4.6 deep eutectic solvents.

Response: Thank you for your comment. We reviewed the number of the section.

In the second paragraph of this section please write “Grudniewska and Pastyrczyk [57,65]”

Response: Response: Thank you for your comment. We reviewed the manuscript according to your suggestion.
